# Comparative Analysis of Influencing Factors on Crash Severity between Super Multi-Lane and Traditional Multi-Lane Freeways Considering Spatial Heterogeneity

**DOI:** 10.3390/ijerph191912779

**Published:** 2022-10-06

**Authors:** Junxiang Zhang, Bo Yu, Yuren Chen, You Kong, Jianqiang Gao

**Affiliations:** 1Key Laboratory of Road and Traffic Engineering of the Ministry of Education, College of Transportation Engineering, Tongji University, 4800 Cao’an Highway, Shanghai 201804, China; 2Engineering Research Center of Road Traffic Safety and Environment, Ministry of Education, Tongji University, Shanghai 201800, China; 3College of Transport and Communications, Shanghai Maritime University, Shanghai 201303, China

**Keywords:** crash severity, super multi-lane freeway, traditional multi-lane freeway, spatial heterogeneity, hierarchical Bayesian approach

## Abstract

With the growth of traffic demand, the number of newly built and renovated super multi-lane freeways (i.e., equal to or more than a two-way ten-lane) is increasing. Compared with traditional multi-lane freeways (i.e., a two-way six-lane or eight-lane), super multi-lane freeways have higher design speeds and more vehicle interweaving movements, which may lead to higher traffic risks. However, current studies mostly focus on the factors that affect crash severity on traditional multi-lane freeways, while little attention is paid to those on super multi-lane freeways. Therefore, this study aims to explore the impacting factors of crash severity on two kinds of freeways and make a comparison with traditional multi-lane freeways. The crash data of the Guangzhou-Shenzhen freeway in China from 2016 to 2019 is used in the study. This freeway contains both super multi-lane and traditional multi-lane road sections, and data on 2455 crashes on two-way ten-lane sections and 13,367 crashes on two-way six-lane sections were obtained for further analysis. Considering the effects of unobserved spatial heterogeneity, a hierarchical Bayesian approach is applied. The results show significant differences that influence the factors of serious crashes between these two kinds of freeways. On both two types of freeways, heavy-vehicle, two-vehicle, and multi-vehicle involvements are more likely to lead to serious crashes. Still, their impact on super multi-lane freeways is much stronger. In addition, for super multi-lane freeways, vehicle-to-facility collisions and rainy weather can result in a high possibility of serious crashes, but their impact on traditional multi-lane freeways are not significant. This study will contribute to understanding the impacting factors of crash severity on super multi-lane freeways and help the future design and safety management of super multi-lane freeways.

## 1. Introduction

With the development of the regional economy and the improvement of car ownership rates, some existing traditional multi-lane freeways (i.e., two-way six-lane or eight-lane) cannot meet traffic capacity needs [1]. To effectively alleviate the contradiction between traffic supply and demand, more and more traditional multi-lane freeways are being renovated and expanded into super multi-lane freeways (i.e., equal to or more than two-way ten-lane) [2]. The world’s super multi-lane freeways are mainly concentrated in developed countries, such as the United States, Germany, France, etc. For example, the United States has at least 43 super multi-lane freeways [3]. In developing countries, such as China, existing super multi-lane freeways are very scarce, and more are under planning and construction. However, current studies on super multi-lane freeways pay little attention to serious crashes, and most of them focus on traffic efficiency, charge management, and so on [4,5,6]. Compared with traditional multi-lane freeways, super multi-lane freeways may bring new risks. Faced with the tide of freeway reconstruction and expansion, ensuring traffic safety and reducing serious crashes on super multi-lane freeways is of great importance.

Super multi-lane freeways have different traffic environments and flow characteristics from traditional multi-lane freeways, which greatly impacts traffic safety. Through the simulation of freeways with different lane numbers, a study has found that the number of lane changes increases with traffic volume and the number of lanes provided for the direction of travel [7]. For freeways with more lanes, lane changes and the interweaving phenomenon of the overall traffic flow are more pronounced, especially at the entrances and exits of freeway interchanges, which can easily lead to high traffic risks [8,9]. In addition, it is well known that freeways with more lanes tend to have higher design speeds in the inner lanes, and extensive research has shown a significant positive correlation between speed and crash severity [10]. It has also been found that although freeways with two-way four lanes have more crashes than those with two-way six lanes, freeways with two-way six lanes have higher rates of fatalities and injuries [11]. A study also reports that there is a significant statistical difference in the crash severity between two-way six-lane (and below) freeways and ten-lane freeways [12]. Thus, serious crashes on multi-lane freeways do not have a simple linear relationship with the increase in the number of lanes, and many complex conditions are worthy of in-depth study.

The factors affecting the severity of crashes on freeways include four main aspects: drivers, vehicles, roads, and the environment. However, most of the studies on this topic are based on crash data from traditional multi-lane freeways and do not distinguish the detailed number of lanes in the analysis. According to general findings from research, the severity of crashes is significantly impacted by driver behavior characteristics, such as speeding, aggressive driving, failure to use seat belts, and driving while ineligible for a license [13]. Through the analysis of two-way four-lane and six-lane freeway crashes, traffic volume, median type, and the number of lanes are found to be significantly associated with fatal injury [14]. In terms of road factors, dry road conditions and wider shoulders can reduce the severity of crashes [15]. In addition, a study on traditional multi-lane freeways reports that vision obstruction is a major factor in serious crashes, and a large percentage of trucks is more likely to cause serious injuries [16].

A series of comprehensive studies are conducted on Florida’s high-speed multi-lane arterial corridors to identify the factors that are responsible for severe and fatal crashes on two-way four-lane to eight-lane freeways [17,18,19]. The result shows that alcohol or drug use is significant across all collision types. Changing lanes in corridors with a large percentage of trucks leads to a high possibility of severe crashes. Poor road conditions and higher speed limits increase the likelihood of serious rear-end crashes. Additionally, failure to use safety equipment can also intensify the severity of crashes.

To analyze the influencing factors of crash severity, previous studies have used many methods, such as binomial or multinomial logistic regression [20], ordered logit or probit models [21], classification trees [22], linear genetic programming [16], etc. Due to the traffic data collection and clustering process, multilevel data structures widely exist. The traditional crash severity model cannot take multilevel data into account, disregarding the possible within-group correlation, which may lead to unreliable parameter estimation and statistical inference of the model [23]. The hierarchical Bayesian approach can address the problems caused by within-group correlations [24]. In addition, the Bayesian inference can update the model using any engineering experiences or justified previous findings as prior knowledge [23,25]. This method can effectively handle missing data, which commonly appear in crash records, by considering the information present in other observed data [26,27]. The hierarchical Bayesian approach that explicitly specifies multilevel structures and reliably generates parameter estimates is recommended [24].

Given the above, there are many studies on the impacting factors of serious crashes on traditional multi-lane freeways. However, little research focuses on super multi-lane freeways. To fill the research gap, this study aims to explore the factors affecting serious crashes on super multi-lane freeways and investigate the differences of these factors between super and traditional multi-lane freeways. A hierarchical Bayesian approach is applied for analysis to avoid biased results arising from unobserved spatial heterogeneity. We expect this study to help provide road design and safety management advice to prevent serious crashes on super multi-lane freeways.

## 2. Materials and Methods

### 2.1. Data Preparation

The data used in this study is from the Guangzhou-Shenzhen freeway in China, which was officially opened in July 1997 with a total length of 119 km. With the increase in traffic demand, some parts of the road were subsequently expanded to two-way ten lanes. Among them, 110.6 km are two-way six-lane, and 8.4 km are two-way ten-lane. The acquired data comprise crashes from January 2016 to March 2019, with a total of 15,822 crashes. A number of 2455 crashes are from two-way ten-lane sections and 13,367 crashes are from two-way six-lane sections. Due to the replacement of the recording data system after April 2019 and the impact of COVID-19 in 2020, the crash data were only obtained until March 2019 in this study. The start and end points and the distribution of different types of sections (i.e., two-way ten-lane and six-lane sections) of the freeway is shown in Figure 1.

In this study, crash severity on traditional and super multi-lane freeways is chosen as the dependent variable. Based on the injury severity of drivers, the crash severity is divided into two levels: serious crashes (i.e., high severity) and slight crashes (i.e., low severity). A crash is considered as a serious crash if at least one of the drivers involved is injured. Otherwise, it is considered as a slight crash. The number of serious and slight crashes per kilometer on both types of freeways is plotted in Figure 2.

According to Figure 2, compared with two-way six-lane freeways, the number of crashes and serious crashes per kilometer on two-way ten-lane freeways is much higher than on two-way six-lane freeways, at 2.42 and 2.19 times that of the two-way six-lane freeway, respectively. Crashes and serious crashes are much denser on two-way ten-lane freeways, so studying the impacting factors of crash severity on super multi-lane freeways is necessary.

The original data records the information of the whole process of crashes from the occurrence to the end of the treatment. We then rejected most of the variables, such as the order number, event notification, and event handling, as these variables are not related to the crash severity as studied in this paper. Trade-offs were also made for highly correlated variables. For a set of highly correlated variables, only one was selected and the rest were removed. Finally, a total of 10 variables categories were used in the crash level classification, containing day of week, time of day, weather, horizontal position, crash cause, type of crash, number of vehicles, heavy vehicle involvement, crash form, and interchange.

The correlation of all independent variables used in models is checked by calculating the variance inflation factors (VIFs) before the models are performed. VIF>10 denotes significant multicollinearity [28,29]. All correlation values are smaller than 10, indicating no significant correlation among all independent variables.

This study analyzes the influence of the location from the horizontal position and the longitudinal position. The horizontal position is classified into three categories: the center lane, the left-most lane, and the right-most lane. The left-most lane is the innermost (left) lane of the freeway for overtaking vehicles and vehicles whose speed meets the requirements. From the perspective of the overall longitudinal route, the interweaving of vehicles in the interchange node area is more obvious than that of ordinary road sections, which may also cause differences in the severity of crashes. Therefore, the longitudinal location of the crash is divided into interchange and non-interchange areas. The interchange area is defined strictly according to the design drawings, and non-interchange areas are defined as the continuous driving sections between two interchanges.

Crash-related features used in this study include crash cause, crash types, number of vehicles, vehicle types, and crash forms. Crash causes include large speed disparity, drivers’ improper operation, etc. The speed disparity is the difference in speed between the colliding vehicle and the surrounding vehicles (e.g., front and rear vehicles, lateral vehicles) before the crash occurs. If the speed difference between vehicles is more than 30 km/h, then it is considered a large speed disparity. Due to the recording methods of the original data, this study divides collision types into four categories, namely, vehicle-to-vehicle collision, vehicle-to-facility collision, vehicle-to-people collision, and rollover. Vehicle-to-facility collision refers to vehicles colliding with transportation facilities, such as medians, guardrails, etc. This study divides the number of vehicles involved in the crash into three categories: single-vehicle crash, two-vehicle crash, and multi-vehicle crash (i.e., more than or equal to three vehicles). Vehicle types are divided into heavy vehicles (i.e., trucks, trailer trucks, and buses) and light vehicles (i.e., light commercial vehicles and passenger cars). The form of crash refers to whether the crash is a single crash (i.e., a crash that occurs alone) or a derivative crash (i.e., a crash that occurs as a result of a previous crash).

The description and classification of the final variables and the statistics of mean and standard deviation are shown in Table 1.

### 2.2. Methodology

This study aims to better understand the factors that influence the severity of crashes on super multi-lane freeways and to compare them with traditional multi-lane freeways. Numerous factors influence crash severity, but the available factors in existing datasets are limited, and the datasets cannot contain all influencing factors. Road geometric characteristics, regional economics, and population conditions can lead to different traffic flows, which may affect the severity of crashes. It is found that the population structure influences road density, and injury-related crashes have a spatial correlation [30]. Considering the unobserved spatial heterogeneity caused by road segments, this study applies a hierarchical Bayesian approach to analyze the impacting factors of crash severity. This approach consists of two parts, the hierarchical model and Bayesian inference.

#### 2.2.1. The Hierarchical Model

Freeway crashes are complex events that involve many factors, such as vehicle factors, roadway features/conditions, traffic flow, and environmental conditions. Once a crash has occurred, more complexities have to be considered, such as impact angles, human characteristics, and vehicle types. It is hard to acquire all of the data that potentially affect the likelihood of serious crashes. The lack of these factors (which constitute unobserved heterogeneity) may lead to biased and inconsistent parameter estimates and erroneous inferences for traditional statistical analysis. The hierarchical model with random parameters can properly consider the potential heterogeneities [31].

Traffic flow conditions (e.g., traffic volume, traffic vehicle mix, and speed limits), lighting conditions, and freeway characteristics (e.g., median barrier, shoulder, pavement conditions, road alignment, and lane width) are different on different road segments, and these potentially unobservable factors lead to the spatial heterogeneity of crashes, which means that there may be within-group correlations for crashes that occur on the same road segment [32]. These influencing factors cannot be obtained, but they can be reflected by road segments. The hierarchical method using a multilevel structure can accurately analyze the influencing factors on crash severity. This research uses a hierarchical binomial logistic model with two levels to analyze the influencing factors. The first level is the crash level, and the second level is the road segment level. Crashes from the same road segment are considered as a cluster. In level 1, the dependent variable $Y_{ij}$ for the ith crash on the jth road segment has only one of two values, 0 and 1. $Y_{ij}=1$ means serious crash (e.g., crash with injury), while $Y_{ij}=0$ means slight crash (e.g., crash without injury). The likelihood of $Y_{ij}$ = 1 is denoted by $\pi_{ij}=\mathrm{Pr}\left(Y_{ij}=1\right)$, which follows a binomial distribution. The function is as follows:(1)$$logit\left(\pi_{ij}\right)=log\left(\frac{\pi_{ij}}{1-\pi_{ij}}\right)=\beta_{0j}+\overset{P}{\underset{p=1}{\sum }}\beta_{pj}X_{pij}+\varepsilon_{ij}$$
where $\beta_{0j}$ is the level 1 (i.e., the crash level) intercept and $\beta_{pj}$ is the estimated regression coefficient for $X_{pij}$. $X_{pij}$ is the value of pth independent variable for ith crash for jth road segment. P is the number of independent variables in level 1, and $\varepsilon_{ij}$ is the disturbance term, which has a mean of zero and variance to be estimated.

The within-crash correlation in level 2 (i.e., the road segment level) is specified as follows:(2)$$\beta_{0j}=\gamma_{00}+\overset{Q}{\underset{q=1}{\sum }}\gamma_{0q}Z_{qj}+\mu_{0j}$$
(3)$$\beta_{pj}=\gamma_{p0}+\overset{Q}{\underset{q=1}{\sum }}\gamma_{pq}Z_{qj}+\mu_{pj}$$
where $\gamma_{00}$ and $\gamma_{p0}$ are estimated intercepts on the road segment level and $Z_{qj}$ is the qth independent variable for jth road segment. $\mu_{0j}$ is the crash-level intercept’s random effect varying across road segments and $\mu_{pj}$ is the random effects of the crash level covariate p. It is assumed that they ($\mu_{0j}$ and $\mu_{pj}$) have normal distributions with means of zero and variances $\sigma_{0}^{2}$ and $\sigma_{k}^{2}$, respectively.

The values of the intercept $\beta_{0j}$ and coefficient $\beta_{pj}$ change with the different road segments and they are determined by two combined components. Since various environmental, road, driver, and vehicle factors may lead to different crash severity, it was first assumed that they have linear relationships with the level 2 covariates $Z_{qj}$. In addition to the fixed components that depend on the level 2 covariates $Z_{qj}$, random effects ($\mu_{0j}$ and $\mu_{pj}$) are also included to account for the unobservable variations between road segments [33]. The random effects only differ between road segments, and they remain constant for all the crashes on the same road segment.

When only the intercept is assumed to be affected by random effects, the covariate component $\overset{Q}{\underset{q=1}{\sum }}\gamma_{pq}Z_{qj}$ of crash level and the random part $\mu_{pj}$ are dropped. Thus $\beta_{pj}=\gamma_{p0}$. The hierarchical model with only the random intercept is shown in Equation (4).
(4)$$logit\left(\pi_{ij}\right)=log\left(\frac{\pi_{ij}}{1-\pi_{ij}}\right)=\gamma_{00}+\overset{Q}{\underset{q=1}{\sum }}\gamma_{0q}Z_{qj}+\overset{P}{\underset{p=1}{\sum }}\gamma_{p0}X_{pij}+\mu_{0j}$$

By substituting Equations (2) and (3) with Equation (1), The full hierarchical model with both random intercept and random slope is demonstrated in Equation (5) [34].
(5)$$\mathrm{logit}\left(\pi_{ij}\right)=\log\left(\frac{\pi_{ij}}{1-\pi_{ij}}\right)\;=\gamma_{00}+\overset{Q}{\underset{q=1}{\sum }}\gamma_{0q}Z_{qj}+\mu_{0j}+\overset{P}{\underset{p=1}{\sum }}\overset{Q}{\underset{q=1}{\sum }}\gamma_{pq}Z_{qj}X_{pij}+\overset{P}{\underset{p=1}{\sum }}\gamma_{p0}X_{pij}+\varepsilon_{ij}$$

The exponential of estimated coefficients is calculated as odds ratios (odds ratio = $e^{\gamma }$) to better interpret the hierarchical Bayesian model. Compared with the reference group, the likelihood of high severity crashes increases or decreases by |$e^{\gamma }$−1| × 100% for the non-reference group.

Since the entire line of the Guangzhou-Shenzhen freeway includes 24 interchanges, and the regional development status, traffic volume, and road conditions are different between these interchanges, the road segments are divided according to the central position of the 24 interchanges in this study.

#### 2.2.2. Bayesian Inference

Bayesian inference is applied to estimate the fixed and random parameters in the hierarchical model. Bayesian inference is a statistical inference method that uses the Bayesian theorem to update the possibility of a hypothesis as more evidence and information becomes available. Bayesian inference derives the posterior distribution as a result of two antecedents: the prior belief and a likelihood function derived from a statistical model of the observed data. This method allows the model to be hierarchical and provides deeper insights using parameter distributions and credible intervals [35,36,37].

Using the Bayesian theorem requires a prior distribution g(π), which is a belief about the potential value of the parameter π before obtaining the data. Three kinds of prior distribution are commonly used: (1) strong informative prior distributions based on the previous investigation or expert knowledge; (2) weak informative prior distribution that does not significantly determine the posterior distribution but can prevent inappropriate inferences; and (3) uniform priors that explain the evidence in the data probabilistically [38]. Uniform priors are used in this study in the absence of prior information. The regression coefficients ($\gamma_{00}$,$\;\gamma_{0q}$, $\gamma_{p0}$, and $\gamma_{pq}$) are assumed as normal distributions (0, 1000). Since the conditional conjugate property of the inverse gamma prior has a more flexible mathematical property [39], the variances $\sigma_{0}^{2}$ and $\sigma_{k}^{2}$ are assumed to obey the distributions of Gamma (0.001, 0.001).

In modern Bayesian statistical inference, the Monte-Carlo (MC) sampling methods developed rapidly, for which the Markov chain Monte-Carlo (MCMC) algorithm is the best known [40]. In the MCMC method, a Markov chain corresponding to the posterior distribution of parameters is needed, which has the same long-term probability as the posterior distribution. The hierarchical Bayesian approach is performed using the package brms in the statistical software R. The brms package uses Stan as its back-end rather than fitting models directly. Therefore, brms models can be fitted using any sampler that has been implemented in Stan. The model in this study is computed by Hamiltonian Monte-Carlo (HMC) sampler, an MCMC technique. Each model initiates two parallel MCMC chains, with 500 initial iterations in each chain dropped as burn-in, and 1000 iterations in each chain are used for generating the descriptive statistics for posterior estimates. The ratio of merge and intra-chain interval width are used to check whether the MCMC chains converge reasonably. If both are around 1, the MCMC chains are thought to converge [41]. The 95% Bayesian credible interval (95% BCI) is used to test the significance of independent variables [42]. If the 95% BCI does not cover 0 (or 95% BCI) of the odds ratio does not cover 1, the parameter estimate is considered to pass the Bayesian credible interval test.

To select the best fitting model, the Watanabe–Akaike Information Criterion (WAIC) [43] and leave-one-out cross-validation (LOO) [44] are used in this study. The model with the lowest WAIC and LOO has the best performance. WAIC is an improvement on the deviance information criterion (DIC). Although DIC has recently gained popularity, it has some significant flaws that are partially caused by the fact that it is only based on point estimation and is not fully Bayesian [44]. For instance, DIC is not defined for singular models and can yield negative estimates of models’ effective number of parameters. WAIC uses the entire posterior distribution, so it is fully Bayesian. A study has shown that it is recommended to try LOO in the limited case of influential observations [45]. Since the crash data from super multi-lane freeways in this study are relatively less than those from traditional multi-lane freeways, LOO is used as a supplementary model evaluation criterion for WAIC in this study.

## 3. Results

### 3.1. Model Comparison

By comparing WAIC and LOO, the best-fit hierarchical Bayesian models are selected for crashes on super multi-lane and traditional multi-lane freeways. The model comparison results are shown in Table 2. For super multi-lane freeways, the hierarchical Bayesian model with only random intercept is the best-fit model, and for traditional multi-lane freeways, the hierarchical Bayesian model with both random intercept and random slope is the best-fit model.

### 3.2. Impacting Factors on Crash Severity of Two Types of Freeways

The result of the best model of crashes on two-way ten-lane freeways is shown in Table 3. For crashes on two-way ten-lane (super multi-lane) freeways, eight variables pass the significance test of 95% BCI. Among these variables, rainy weather, left-most lane, vehicle-to-facility, two-vehicle, multi-vehicle, and heavy vehicle all positively affect crash severity, whereas daytime and peak time are negatively correlated with crash severity.

This study also analyzes the factors that affect serious crashes on two-way six-lane freeways in order to make a comparison with two-way ten-lane freeways. Seven variables have passed the significance test of 95% BCI in the final Bayesian model. The results show in Table 4 that two-vehicle, multi-vehicle, heavy vehicle, interchange, and left-most lane all have positive effects on crash severity, while daytime and peak time are negatively associated with crash severity.

## 4. Discussion

This study compares the influencing factors of super multi-lane freeways with traditional multi-lane freeways to explore the characteristics of influencing factors of super multi-lane freeways. The odds ratios of the best-fit models for both types of freeways are presented in Figure 3 to intuitively demonstrate the results.

### 4.1. The Environment of Crashes

Time has a significant impact on the severity of crashes on both super multi-lane and traditional multi-lane freeways. For super multi-lane freeways, daytime (OR = 0.70) crashes are 30% less likely to be serious than those that occur at night. For traditional multi-lane freeways, compared to nighttime crashes, daytime crashes (OR = 0.73) are 27% less likely to be serious. In other words, crashes at night have a higher possibility of being serious. Due to the poor visibility of driving at night, the driver cannot easily observe surrounding road conditions and cannot brake in time when approaching a vehicle or obstacle ahead [46]. It has been widely suggested that upgrading road lighting can greatly improve freeway safety, especially at night [34]. Another potential rationale could be that the injured driver may not receive timely treatment at night, making the injury worse. The effect of nighttime is more pronounced on super multi-lane freeways, which might be due to the higher speed of vehicles on these roads.

Compared to off-peak crashes, the possibility of high severity decreases during peak time by 21% on super multi-lane (OR = 0.79) and by 18% on traditional multi-lane (OR = 0.82) freeways, respectively. It can be inferred that due to the higher traffic flow, speed during peak hours is significantly lower than that during non-peak hours, thus reducing the severity of crashes [47]. A study shows that the chances of a serious crash during peak times are even reduced by 60% at intersections [34]. For super multi-lane freeways, the influence of peak time on crash severity is a little smaller than that on traditional multi-lane freeways. This may be because traditional multi-lane freeways cannot meet traffic demand and are at near saturation during both off-peak and peak times, with no significant difference in speed [48].

In addition, rainy weather has a significant positive effect on crash severity on super multi-lane highways but not on traditional multi-lane highways. There is an increase of 79% in the likelihood of crash severity if the weather is rainy on super multi-lane highways (OR = 1.79). Visibility is reduced during rain, which may result in drivers’ not being able to detect vehicles or obstacles ahead in time, and rain makes the road more slippery and increases vehicles’ braking distances [49,50]. The effects of rain may be more significant on super multi-lane highways due to the higher traffic volumes and higher design speeds.

### 4.2. The Location of Crashes

The cross-section location of the crash is related to whether the crash is serious. Crashes that occur in the left-most lane are more likely to be serious for both the super multi-lane and traditional multi-lane freeways. For super multi-lane freeways, the left-most lane raises the chance of high crash severity by 29% (OR = 1.29) in comparison to the center lanes. For traditional multi-lane freeways, the probability of high severity increases by 21% (OR = 1.21) in crashes occurring on the left-most lane. The speed of the vehicle is higher in the left-most lane, and it usually rises with the increase in the number of freeway lanes. The relationship between vehicle speed and crash severity has been demonstrated as positive [47].

For traditional multi-lane freeways, serious crashes are more likely to occur in the interchange zone, which has a 40% increased likelihood of high severity (OR = 1.40). However, for super multi-lane freeways, the location of crashes has no significant relationship with high severity. It has been shown that different levels of interchange can affect the severity of crashes [51]. The interchange area of super multi-lane freeways has a stronger function of converting traffic flow, and its alignment design of ramps and diverging and merging areas may have higher standards, allowing vehicles to run more smoothly, which makes it less likely to become a serious crash-prone location compared with the interchange area of traditional multi-lane freeways.

### 4.3. The Involved Vehicles in Crashes

When heavy vehicles are involved, high crash severity is more probable. The likelihood increases by 146% on traditional multi-lane freeways (OR = 2.46), while on super multi-lane freeways, there is an increase of 213% (OR = 3.13), demonstrating a greater positive impact. According to a study by Ahmed et al., large trucks are 2.3 and 4.5 times more likely to result in serious crashes than other vehicles on state and interstate routes, respectively [52]. Heavy vehicles are much larger, and drivers can easily have more blind spots, where they cannot see other vehicles [53]. In addition, they have more weight and greater impact force in crashes, especially when traveling at higher speeds [51]. The influence of heavy vehicles deserves significant attention.

Two-vehicle involvements increase the severity of crashes compared to single-vehicle collisions on both types of freeways. For super multi-lane freeways, compared with single-vehicle collisions, the possibility of high crash severity raises by 194% (OR = 2.94) if two vehicles are involved, and for traditional multi-lane freeways, the likelihood of high severity increases by 183% (OR = 2.83). Multi-vehicle involvements also increase the severity of crashes on both types of freeways, and the influencing degrees increase rapidly. A positive dependence has been found between crash size (i.e., number of vehicles involved) and crash severity [54]. It can be considered that the more vehicles involved in a crash, the greater the likelihood of driver exposure and injury. Compared to single-vehicle crashes, the likelihood of high severity increases by 733% (OR = 8.33) on super multi-lane freeways and by 669% (OR = 7.69) on traditional multi-lane freeways. Compared to traditional multi-lane freeways, the impact of two-vehicle and multi-vehicle involvement on the super multi-lane freeways is greater. Super multi-lane freeways have more lanes, which may involve more vehicles when a crash occurs, and they are designed for vehicles with higher speeds, which may result in higher severity when a multi-vehicle crash occurs [55]. In addition, super multi-lane freeways have better pavement design standards and better road conditions. Some studies have found that highly serious crashes are expected to occur when drivers drive on roads with good road conditions because drivers tend to be overly relaxed and engage in risky driving, which leads to more serious crashes [56,57].

### 4.4. The Type of Crashes

For super multi-lane freeways, collisions with facilities are significantly positively correlated with high severity. The likelihood of high severity increased by 144% (OR = 2.44). This means that collisions with transportation infrastructure carry a higher risk of severe injury or death than collisions between vehicles. It was also found that fixed roadside objects can affect crash severity [58]. The reason for the different severity levels caused by crash types may be partly from the difference in relative speed, as the relative speed of a collision with a traffic facility may be greater compared to a collision with a vehicle in the same direction, which leads to a more serious crash. In addition, traffic facilities are made of harder materials and have less deformation to buffer the collision.

## 5. Conclusions

This study explores the factors affecting the crash severity of super multi-lane freeways, and then makes comparisons with traditional multi-lane freeways. To take the unobserved spatial heterogeneity and intra-class correlations of crashes into account, a hierarchical Bayesian approach is applied for analyses. The results show that there are significant differences in influencing factors between the two kinds of freeways.

Although some factors significantly affect crash severity on both types of freeways, the degrees of influence are different. For instance, heavy-vehicle, two-vehicle, and multi-vehicle involvements are more likely to lead to serious crashes on both super multi-lane and traditional multi-lane freeways but the degrees of impact of these factors on super multi-lane freeways are much higher. For super multi-lane freeways, in addition to the above three factors, vehicle-to-facility collisions increase the possibility of high severity compared with vehicle-to-vehicle collisions. Rainy weather also increases the possibility of serious crashes.

With the growth of the transportation demand, super multi-lane freeways are the development trend of future freeways. Based on the findings of this study, several management measures are recommended to reduce the crash severity of super multi-lane freeways. For example, lighting conditions should be improved at night. Timely reminders and appropriate speed limits should be applied in bad weather, especially in rainy weather. Since multi-vehicle crashes have a high probability of causing injury or death, informational cues and guidance need to be enhanced to reduce the likelihood of multi-vehicle crashes, especially in adverse conditions (e.g., bad road conditions, limited vision, etc.). Heavy vehicle involvement is also a significant factor affecting serious traffic crashes, so special, separate lanes for heavy vehicles are strongly recommended to reduce unnecessary interactions between heavy vehicles and light passenger cars. Blind spot warnings for heavy vehicles are also necessary. Moreover, to reduce the severity of vehicle–facility crashes, buffer and anti-collision materials are advised to be used in traffic facilities (e.g., median dividers, guardrails, etc.).

This study has some limitations that should be addressed in future work. It has been shown that the coupling of freeway lane-types and traffic conditions will affect the severity of crashes [59,60]. Therefore, other important factors, such as real-time traffic conditions, area types, road alignments, and driver characteristics, need to be included in further analyses.

## Figures and Tables

**Figure 1 ijerph-19-12779-f001:**
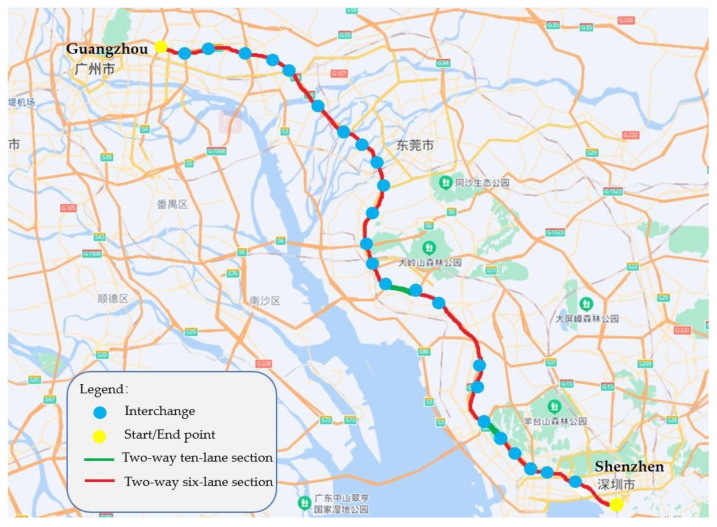
The Guangzhou-Shenzhen freeway.

**Figure 2 ijerph-19-12779-f002:**
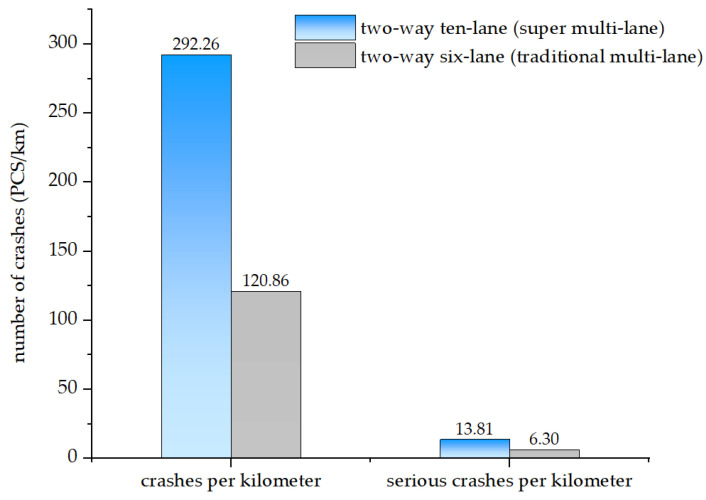
Comparison of crashes per kilometer between two-way ten-lane and six-lane freeways.

**Figure 3 ijerph-19-12779-f003:**
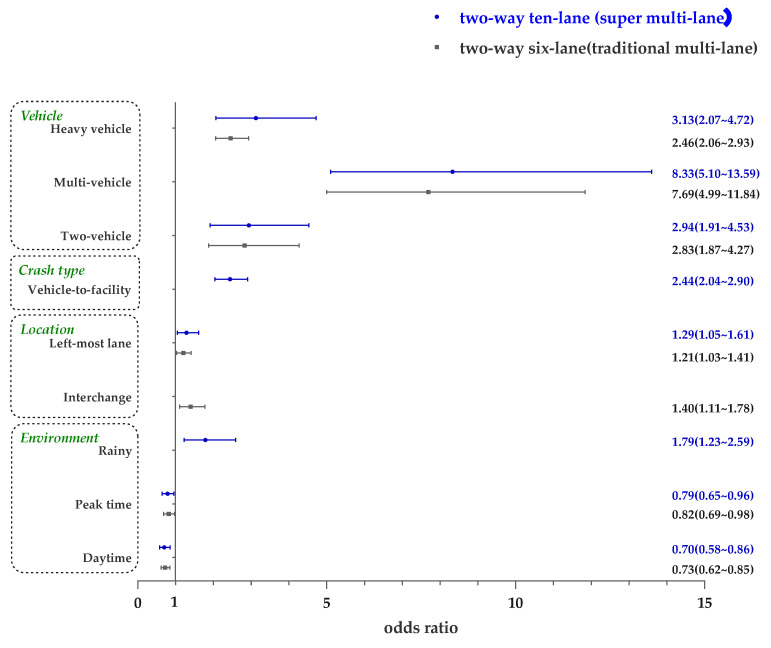
Odds Ratio of the influencing factors for crash severity on two-way ten-lane and six-lane freeways.

**Table 1 ijerph-19-12779-t001:** Descriptive statistics of crashes.

Variable Name	Description	Percent	
		Two-Way Ten-Lane	Two-Way Six-Lane
Dependent variable			
Slight crash	If the crash is slight = 0	95.35%	94.82%
Serious crash	If the crash is serious = 1	4.65%	5.18%
Independent variables			
Day of week	If crash at workday = 1, otherwise = 0	66.73%	67.14%
Time of day			
Day time	If crash at 7 a.m.–6 p.m. = 1, otherwise = 0	69.71%	65.18%
Peak time	If crash at 7 a.m.–10 a.m. or 5 p.m.–8 p.m. = 1, otherwise = 0	24.34%	28.91%
Weather			
Sunny *	If weather was sunny = 1, otherwise = 0	94.65%	93.73%
Rainy	If weather was rainy = 1, otherwise = 0	2.92%	3.83%
Cloudy	If weather was cloudy = 1, otherwise = 0	2.43%	2.44%
Horizontal position			
Center lane *	If crash on center lane = 1, otherwise = 0	35.62%	33.83%
Left-most lane	If crash on over-taking lane = 1, otherwise = 0	44.46%	57.35%
Right-most lane	If crash on shoulder = 1, otherwise = 0	19.92%	8.82%
Crash cause			
Speed disparity *	If the speed disparity is large = 1, otherwise = 0	94.58%	91.02%
Improper operation	If there is driver improper operation = 1, otherwise = 0	3.84%	7.09%
Unknown	If crash cause is unknown = 1, otherwise = 0	1.58%	1.89%
Type of crash			
Vehicle-to-vehicle *	If crash is vehicle-to-vehicle = 1, otherwise = 0	94.59%	91.03%
Vehicle-to-facility	If crash is vehicle-to-facility = 1, otherwise = 0	3.32%	5.78%
Vehicle-to-people	If crash is vehicle-to-people = 1, otherwise = 0	0.22%	0.25%
Rollover	If crash is rollover = 1, otherwise = 0	1.87%	2.94%
Number of vehicles			
Single-vehicle *	If single-vehicle crash = 1, otherwise = 0	5.65%	8.92%
Two-vehicle	If Two-vehicles crash = 1, otherwise = 0	64.87%	60.51%
Multi-vehicle	If multi-vehicle crash = 1, otherwise = 0	29.48%	30.57%
Heavy vehicle involvement	If heavy vehicle involved = 1, otherwise = 0	15.53%	15.28%
Crash form	If it is non-consecutive crash = 1, otherwise = 0	95.61%	94.63%
Interchange	If crash at interchange = 1, otherwise = 0	29.51%	12.16%

Note: * denotes the reference class of polynomial variables used in the model.

**Table 2 ijerph-19-12779-t002:** WAIC and LOO of hierarchical Bayesian models with different structures.

	Bayesian Logistic Regression Models (with Only Fixed Effects)		Hierarchical Bayesian Models with Random Intercept		Hierarchical Bayesian Models with both Random Intercept and Random Slope	
	WAIC	LOO	WAIC	LOO	WAIC	LOO
Models for crash severity on super multi-lane highways	899.21	899.62	885.34	886.07	897.21	897.32
Models for crash severity on traditional multi-lane highways	5147.84	5139.57	5088.75	5088.83	5031.12	5030.63

**Table 3 ijerph-19-12779-t003:** Estimation results for crashes on two-way ten-lane freeways.

Parameters	Estimate (Std Err)	Odds Ratio (95% Confidence Interval)
Fixed parameters		
Time of day		
Day time	−0.35 (0.10)	0.70 (0.58~0.86)
Nighttime *	0	1
Peak time	−0.24 (0.11)	0.79 (0.65~0.96)
Off-peak time *	0	1
Weather		
Rainy	0.58 (0.19)	1.79 (1.23~2.59)
Sunny *	0	1
Horizontal position		
Left-most lane	0.26 (0.11)	1.29 (1.05~1.61)
Center lane *	0	1
Type of crash		
Vehicle-to-facility	0.89 (0.09)	2.44 (2.04~2.90)
Vehicle-to-vehicle *	0	1
Number of vehicles		
Two-vehicles	1.08 (0.22)	2.94 (1.91~4.53)
Multi-vehicles	2.12 (0.25)	8.33 (5.10~13.59)
Single-vehicle *	0	1
Heavy vehicle involvement		
Heavy vehicle	1.14 (0.21)	3.13 (2.07~4.72)
Non-heavy vehicle *	0	1
Intercept (level 1)	−2.66 (0.32)	0.07 (0.04~0.12)
Random parameters		
Intercept (Segments)	0.80 (0.41)	2.23 (1.02~25.02)

Note: * represents the reference class of polynomial variables used in the model.

**Table 4 ijerph-19-12779-t004:** Estimation results for crashes on two-way six-lane freeways.

Parameters	Estimate (Std Err)	Odds Ratio (95% Confidence Interval)
Fixed parameters		
Time of day		
Day time	−0.32 (0.08)	0.73 (0.62~0.85)
Nighttime *	0	1
Peak time	−0.20 (0.09)	0.82 (0.69~0.98)
Off-peak time *	0	1
Interchange	0.34 (0.12)	1.40 (1.11~1.78)
Non-interchange *	0	1
Horizontal position		
Left-most lane	0.19 (0.08)	1.21 (1.03~1.41)
Center lane *	0	1
Number of vehicles		
Two-vehicle	1.04 (0.21)	2.83 (1.87~4.27)
Multi-vehicle	2.04 (0.22)	7.69 (4.99~11.84)
Single-vehicle *	0	1
Heavy vehicle involvement		
Heavy vehicle	0.90 (0.09)	2.46 (2.06~2.93)
Non-heavy vehicle *	0	1
Intercept (level 1)	−2.49 (0.16)	0.08 (0.06~0.11)
Random parameters		
Peak time	−0.12 (0.08)	0.88 (0.62~0.98)
Heavy vehicle	1.02 (0.14)	2.77 (2.23~3.17)
Intercept (Segments)	0.68 (0.51)	1.97 (1.24~3.99)

Note: * represents the reference class of polynomial variables used in the model.

## Data Availability

The data presented in this study are available on request from the corresponding author due to privacy.

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
