# Peer review of "Comparative Analysis of Influencing Factors on Crash Severity between Super Multi-Lane and Traditional Multi-Lane Freeways Considering Spatial Heterogeneity"

_ijerph, 2022, doi:10.3390/ijerph191912779_

Round 1
Reviewer 1 Report
The study established a Hierarchical Bayesian model to explore the impacting factors of crash severity on super multi-lane highways considering spatial heterogeneity and then made a comparison with those on traditional multi-lane highways. Given the trend of highway reconstruction and expansion, this study is important and interesting with its unique contributions, especially for some experts and researchers in highway design and safety management. The method and analysis of this study are well-organized and self-justified. Before it is accepted, I have some minor suggestions to improve this study, as follows:
(1) In Line 154, this study divided crashes into two different types, consecutive and non-consecutive crashes. Are the definitions of non-consecutive and consecutive crashes the same as single and derivative crashes mentioned in this study?
(2) In Table 1, what’s the definition of speed disparity? What’s the threshold of large speed disparity?
(3) It had better illustrate two-way ten-lane and six-lane road sections in Figure 2.
(4) In Lines 318-323, the authors argue that the reason why time factors (including daytime and peak time) have greater impacts on crash severity of super multi-lane highways than that of traditional ones might be traffic congestion. However, traffic congestion may reduce vehicle speed with a lower possibility of serious crashes. Traditional multi-lane highways are more likely to be congested than super multi-lane highways, which may make time factors have bigger effects on traditional multi-lane highways. Thus, the current explanation may not be convincing enough. Please further explain the possible reasons.
(5) In Lines 365-369, please add more explanations on the different impacts of two-vehicle and multi-vehicle involvements on crash severity, especially on super multi-lane highways.
Reviewer 2 Report
1, I'm a little bit coufused about the research object in this research, highways or freeways? The authors may clarify that.
2, At line 15 in the abstract, "To fill the research gap" seems like an inappropriate statement
3,Is there any difference between the problem (crash influencing factors on super multi-lane freeways) proposed by the authors and the crash risk analysis on normal freeways? Can the data used explain the practical differences between the two types of freeways? Is Hierarchical Bayes able to address the problem of spatial heterogeneity and the deviation caused by missing data?
4,The universality of research conclusions is doubtful.
5,Abstract should be the summery of the paper. Abstract should write more specific format, not only the details of the background. Abstracts should contain following common elements: Problem: Describe the major topic or problem addressed in the document. Method: Describe the specific approach or method used to solve the problem. Results: Write most important results. Conclusion: Describe the conclusion drawn from the results.
It seems like the abstract misses more general conclusions and not concrete results given as the example, results in the abstract part should include and be expressed with the main data from your results.
6,Future work needs more discussions as using the views of freeway lane-types and the application of this theory in solving traffic crash severity problems, the following references may be cited, e.g.,
Identification of dynamic traffic crash risk for cross-area freeways based on statistical and machine learning methods[J]. Physica A: Statistical Mechanics and its Applications, 595 (2022): 127083.
8,It seems that “super multi lane" is not a professional vocabulary in the field of traffic engineering . You may use a more appropriate word to replace it.。
9,Last but not the least, English proof reading is very important for this paper, the authors should have polished the manuscript carefully before submission or ask for the help from a native English speaker.
10,The applicability of the hierarchical binomial logistic model with two levels to your research data and questions should be explained in more detail.
11,Formulas 1-5 seem to be basic knowledge, which can be replaced by appendixes or references.
12,I suggest to put figure2 in the data description section.
Round 2
Reviewer 2 Report
I have no futrue concerns, it is qualified for publication.